# Active Video Games Performance and Heart Rate on the Wii or Kinect in Children with and without Developmental Coordination Disorder

**DOI:** 10.3390/children9121823

**Published:** 2022-11-25

**Authors:** Jorge Lopes Cavalcante-Neto, Dorothee Jelsma, Tatiane Targino Gomes Draghi, Eloisa Tudella, Bouwien Smits-Engelsman

**Affiliations:** 1Department of Human Sciences, State University of Bahia, Jacobina 44700-000, Brazil; 2Clinical and Developmental Neuropsychology, University of Groningen, 9712 TS Groningen, The Netherlands; 3Department of Physical Therapy, Federal University of Sao Carlos, Sao Paulo 13565-905, Brazil; 4Department of Health and Rehabilitation Sciences, University of Cape Town, Cape Town 7701, South Africa

**Keywords:** active video games, developmental coordination disorder, motor learning, peak heart rate, training intensity

## Abstract

Our objective was to compare changes in game performance and intensity of heart rate (HR) between two types of active video game (AVG) in children with and without Developmental Coordination Disorder (DCD). Additionally, we assessed the level of improvement per game as well as the perceived exertion and enjoyment during training. Seventy-six children, 36 with DCD and 40 without (TD) were randomly assigned to a 5-week program of Wii-Fit or Xbox-Kinect training 2× a week. The steepness of the performance curves was not different between consoles, nor between groups. Playing Kinect games resulted in higher HR in both groups. Wii and Kinect seem to be comparable AVG consoles that can be used for children with and without DCD, with the Kinect reaching a higher intensity of training.

## 1. Introduction

Children with developmental coordination disorder (DCD) display particularly inadequate motor skill acquisition. They represent a specific motor impairment group, with no associated neurological or medical disability, but with constraints in daily activities and sports [1,2]. Active video games (AVGs) or exergames are known to promote gains in motor performance for children with DCD [1,2] and it is assumed that the augmented feedback given by the AVGs may improve motor skills acquisition.

AVGs are resources of technology that deliver augmented feedback, mostly visual information for each movement the gamer makes [3]. The sophisticated interfaces register body movements by sensors that capture small or large body shifts [4]. Straker and colleagues [5] emphasize four key features of AVGs needed to reach a large part of the population: (1)—the acquisition of the hardware and software should be affordable; (2)—it should be clear how to play the games properly, which will promote learning; (3)—the players should be motivated to continue playing the games regularly, which ensures the consistency of the practice, since only habitual use will determine changes in functional outcomes; (4)—sufficient body movements are necessary to become successful in the games due to appropriate movement.

Commercial AVGs or games based on virtual reality technology are generally accessible with considerable low costs [6], such as the Nintendo Wii Fit (Nintendo, Kyoto, Japan) or Xbox Kinect (Microsoft, Redmond, Washington, United States). The Nintendo Wii Fit makes use of a remote controller and Wii balance board to actively steer the virtual character by movements of the hands and shifts of weight while standing on the board [7]. On the contrary, the Xbox Kinect makes use of a sensor integrated with infrared camera that captures the body movements in three dimensions [8] to steer the virtual character and play the game without any additional restrictions, such as a remote controller or balance board [9]. Both types of consoles are feasible and promising as a rehabilitation approach for enhancing physical activity in children with motor impairments [10,11,12,13,14], including those with DCD [15,16,17].

Due to the interactive gaming platform and augmented feedback, these active video games can be used to study gains in motor performance for children with DCD. Some investigations have attested significant improvements in motor skills after Wii Fit-based training in children with DCD [18,19,20,21], but a direct comparison of the effects between different consoles during training has not been carried out yet. So far, the results have attested similar benefits for both intervention types [18,19,20,21], and show task-specific effectiveness, but seem to depend on the intensity, duration and variability principles of training [22]. Most importantly, it is known that children enjoyed this type of training over a six-week period of time [22]. Besides the level of enjoyment, the level of exertion may also be a factor that can help children with DCD to remain motivated when interventions extend over a longer period of time. It is also known in adults that minute ventilation, heart rate, oxygen consumption and energy expenditure are all increased by playing Nintendo Wii boxing, XBOX Kinect boxing and Sony PlayStation (Sony, Konan, Tokyo, Japan) Move gladiatorial combat, and the presence or absence of a controller or camera has no influence on the amount of physical activity required to play the AVGs [18,19,20,21]. However, contradictorily, these game systems seem not have met a moderate level of physical active intensity except for one game of the Kinect (“Your Shape Fitness Evolved 2012”) that reached the intensity of aerobic physical activity recommended for health benefits in adults [18,19,20,21]. Apparently, the type of game seems to differentially enhance certain effects. It is still unknown which console is the best option for children in order to achieve motor learning and recommended training intensity over sessions.

Thus, the objective of this study is to compare changes in game performance and game intensity based on HR effects during two types of AVG training, Wii Fit and Xbox Kinect, in children with and without DCD, both on group and individual level. Additionally, the level of improvement per game as well as the perceived exertion and enjoyment during training were assessed.

## 2. Materials and Methods

### 2.1. Study Design

This randomized comparator-controlled trial was approved by the local Human Research Ethics Committee, under number 89993118.8.000.5504/2018. Informed consent was signed by the parents and assent was given by the children before data collection.

### 2.2. Participants

Children (7–12 years) from two public schools of Sao Carlos, Sao Paulo, Brazil participated in this study. In both schools, parents received a sealed envelope through their children, containing a letter explaining the procedure and two questionnaires (Developmental Coordination Disorder Questionnaire—DCDQ and a demographic questionnaire) as well as the Informed Consent Form. Based on the DCDQ cut-off scores (≤46 from 5 to 7 years and 11 months; ≤55 from 8 to 9 years and 11 months; and ≤57 from 10 to 15 years old) [23], children with reported motor problems were invited to participate in the study just like the children from the same classes whose parents reported no motor problems. The DSM-5 criteria were used to determine classification of DCD [1,2]. All children who scored a total test score of ≤5th percentile on the Movement Assessment Battery for children, second edition (MABC-2) and whose parents indicated poor motor performance interfering in daily or sports life according to the DCDQ, through the sociodemographic questionnaire an early onset of poor motor performance, with no presence of any disability, neurological disorder or co-occurrences as Attention Deficit Hyperactivity Disorder (ADHD) were included as children with DCD. Children from the same classes were classified as typical development (TD) when they scored >7th TSS on the MABC-2 and their parents indicated no problems in motor performance.

We calculated the sample size of this study through G* Power v3.1.9.2 software (Germany), using as reference previous studies [16,24] for repeated measure (pre-post) with two between subject groups (Wii x Kinect; DCD x TD children). We assumed a power of 0.90, an alpha of 0.05 and an effect size of 0.66. The total minimal sample size was estimated in 64 children, 16 per group to detect differences in motor learning and heart rate variability outcomes of this study.

### 2.3. Instruments

Movement Assessment Battery for Children, second edition (MABC-2)—The MABC-2 is a motor performance test with eight items in three domains (manual dexterity; aiming & catching; balance) and has proven to be a validated and reliable tool for Brazilian children [25,26]. Raw scores are converted into standard scores (SS) per domain and a TSS based on age-referenced norms. Motor performance scores are classified as follows: ≤5th TSS significant motor problems; 6–7 TSS at risk for motor problems; and >7th TSS no motor problems [26]. The concurrent validity of the MABC2 is good [27,28] and inter-rater reliability high [26,29].

Developmental Coordination Disorder Questionnaire (DCDQ)—A Brazilian version of the DCDQ was used to screen signs of motor problems in children [23]. The DCDQ consists of 15 items on a 5-point Likert scale. The total score can distinguish between ‘probable DCD’ (≤46 for 5–8 years; ≤55 for 8–9 years; ≤57 for 10–15 years or ‘probably not DCD’.

Parents’ questionnaire—A demographic questionnaire was used to collect information regarding age, gender, history of any concern during pregnancy, at birth or throughout development to check the possible onset of the symptoms related to DCD and the presence of any medical, neurological condition or co-occurrences as ADHD.

Nintendo Wii-Fit—The Nintendo Wii-Fit is a commercial active video game system, with a console connected to a remote controller and a balance board via Bluetooth. In the corners of the balance board, four plate sensors measure the child’s weight and weight distribution to calculate the center of pressure (CoP). The children can use the remote controller by pressing the button, while whole-body movements are detected as CoP displacements resulting in the movements of the avatar during the games. Ten balance games were used to monitor the changes in performance during training. At the end of each game session, the achieved score was shown on the screen and registered by the therapists. The scores were based on the number of correct responses, distance and/or time.

Xbox Kinect—The Xbox Kinect (Microsoft Corp) consists of an Xbox Kinect video game and an infrared camera, which acts as a 3D sensor to detect whole-body movements. This system provides a controller-free type of gaming in which the child uses body movements to control the avatar of the game. Nine games of the adventure and sport genre were used in this protocol. Similarly to the Wii-Fit, the game scores were shown on the screen and registered by the therapists. The scores were based on number of correct responses, sequence and/or time.

Cardio frequency meter—The heart rate (HR) of the children was monitored at rest before the training and during the training sessions, using the Polar V800 cardio frequency meter cooperating with the transmitter placed with a flexible chest strap. In total, HR was monitored for about 30 min at first, fifth and last training session, and the minimum, medium and peak values were extracted on the computer.

The OMNI scale [30]—The OMNI Rating of Perceived Exertion scale with pictorial descriptors on a range of 0 (not tired at all) until 10 (very, very tired) was used to measure the perceived exertion after each training. The Children’s OMNI scale of perceived exertion (OMNI Scale) has been validated for children, attesting good correlation values (0.85–0.94) in relation to the total body oxygen uptake [30].

Enjoyment scale—The enjoyment scale with a range of 0 (Very boring) to 4 (Awesome) was used to measure the amount of enjoyment after each training and was developed for our earlier studies [16].

### 2.4. Randomization

In total, 80 children were randomly assigned to the Wii-training or Kinect training group by a blinded assessor per DCD (*n* = 40) and TD (*n* = 40) group. However, four children with DCD assigned to the Wii group were lost to follow up after the randomization process, because one child moved to another area, one child presented behavioral problems and two children fell ill for a longer period of time and missed too many training sessions. Therefore, 76 children were used for the analyses, with 36 children in the Wii group (16 DCD; 20 TD group) and 40 children in the Kinect group.

### 2.5. Training and Procedure

One week before the training started, all researchers involved in the study tested the feasibility of the protocol by performing an intensive 12 h training session regarding the task rules, duration, time of sessions and scores of each device. Doubts and disagreements were solved by consensus. The training was conducted in specific rooms at the children’s schools, using four televisions (two for Wii and two for Kinect) separated by cardboard. Over nine sessions, children of both groups received 20 min time-on-task training sessions twice a week. Two trained therapists supervised four children who played simultaneously and registered the scores obtained by the children in each session. In the Wii training, children played 10 games twice (‘Soccer heading’, ‘Table tilt’, ‘Ski jump’, ‘Balance bubble’, ‘Penguin slide’, ‘Snowboard slalom’, ‘Kung fu, ‘Obstacle course’, ‘Skate boarding’ and ‘Perfect 10′), spread over two days per week. In the Kinect training, children played nine games (‘River rush’, ‘Rally ball’, ’20,000 Leaks’, ‘Reflex ridge’, ‘Space pop’, ‘Track and field’, ‘Football’, ‘Beach volleyball’ and ‘Table tennis’), with the first five games on the first training day and the last four on the second day of the week. Children played each one of the first five games twice, while each one of the last four games were played only once due to the length of the game.

### 2.6. Data Analysis

To check for differences between the AVG groups and DCD and TD group, demographic characteristics and motor test outcomes were evaluated at baseline using Chi squared test (gender) or t-tests (age, BMI, MABC-2 total score). An equivalent aggregate score was created to be able to test differences in improvement in the different game performances during training. The best score of the first five games per week was used for the analyses of progress in performance over five weeks. Differences between the Wii and Kinect were tested with the t-test analysis and with regard to the game performance outcomes, because there are five moments, for any of the five outcomes to be considered statistically significant, we decided to keep a significance level of <0.1. The other games were regarded as training. All game scores were normalized, after eliminating outliers, as the percentage of the maximum score obtained for that game over the whole training period [23], with a higher normalized game score representing better performance. Afterwards, transformation data were checked for normality. Differences between the Wii and Kinect and subsequently, differences between the TD and DCD group were tested with t-test analysis. To estimate the rate of change on the Normalized Game score, a linear curve was fitted to the data points for each child through regression curve estimation. The steepness of the slopes (β) reflects the extent to which performance improves as training progresses and was calculated over the total score of the five games and per game over the five weeks in which these games were played. Estimates of slope (β) per child were tested with a t-test between AVG groups and sequentially within groups (DCD, TD) between AVG groups. Slopes were also used to classify improvement as (0) not significant; (1) trend towards significant improvement between 5–10% level; (2) significant ≤5% level and (3) significant ≤1%. Differences in classification per console are depicted using cumulative bar carts and frequencies per improvement classification were tested using the Chi squared test between AVG groups and between children with and without DCD.

Mean rest HR and peak HR were calculated per time moment (week 1, 3, and 5) and as the mean over the three recorded training occasions. The estimated maximal heart rate (EMHR) and number of children that reached 60, 70 and 80% of the EMHR was calculated to estimate training intensity. The mean OMNI- and frequencies of the enjoyment scores were tested for differences between the AVG groups and between the children with and without DCD by t-tests or Chi-squared tests. All statistical analyses were performed using the Statistical Package for the Social Sciences (SPSS Inc., Chicago, IL, USA, version 27).

## 3. Results

### 3.1. Group Comparability

No differences were found between the Wii or Kinect groups for age (*t*(74) = −0.35, *p* = 0.726; mean Wii group 9.8 years (SD 1.3), mean Kinect group 9.7 years (SD 1.0)), weight (*t*(74) = −0.06, *p* = 0.953, height (*t*(74) = 0.25, *p* = 0.806) nor MABC total standard score (*t*(74) = −0.17, *p* = 0.863; mean Wii group 7.31 (2.9); mean Kinect group 7.43 (3.1)). Gender distribution was not different between the Wii and Kinect groups (*χ^2^* = 0.84, *p* = 0.358). Within the AVG groups, no differences in age, weight, height and MABC2 total scores were found between the Wii DCD and Kinect DCD group (all *p* > 0.654). Likewise, no differences were found between the Wii TD and Kinect TD group (all *p* > 0.413). For details, see Table 1.

### 3.2. Performance in Percentage of Normalized Game Scores during the 5 Weeks of Training

Overall, a significant higher percentage of normalized game score was found in the Kinect group compared to the Wii, which was present every week (*t*(74) = −11.48, *p* < 0.001; *t*(74) = −15.65, *p* < 0.001; *t*(74) = −17.28, *p* < 0.001; *t*(74) = −19.36, *p* < 0.001; *t*(74) = −17.28, *p* < 0.001, respectively). This difference between Wii and Kinect percentages of maximum scores was present within the TD group and likewise within the DCD group (all *p* < 0.001). No differences were found in normalized game score between the TD and DCD group (Week 1 *t*(74) = 2.05, *p* = 0.044), (Week 2 *t*(74) = 0.48, *p* = 0.636; Week 3 *t*(74) = 0.64, *p =* 0.522; Week 4 *t*(74) = 1.30, *p* = 0.199; Week 5 *t*(74) = 1.11, *p* = 0.272) (Figure 1). The steepness of the improvement (slope β) was not different between consoles (*t*(74) = 0.61, *p* = 0.545; mean Wii β = 0.15 (SD 0.13); mean Kinect β = 0.14 (SD 0.05)) and not between the TD group and DCD group (*t*(74) = −0.45, *p* = 0.652; mean TD β = 0.14 (SD 0.11); mean DCD β = 0.15 (SD 0.07).

On further exploration of the individual classification of significant improvement curves, no differences were found between consoles (χ2 = 3.31, *p* = 0.347) nor between the TD groups that trained either on the Wii or on the Kinect (χ2 =5.99, *p* = 0.112), nor between the DCD groups (χ2 = 4.37, *p* = 0.225) (Figure 2).

As shown in Figure 3A,B, the TD children improved most in the first game of the Wii (Soccer heading) whereupon game 3 (Ski jump) and 4 (Balance bubble) were easiest to improve in. On the Kinect, the first (River rush), third (20,000 leaks), fourth (Reflex ridge) and fifth game (Space pop) were easiest to improve in. For the children with DCD, the first game of the Wii was also the easiest to improve in, with the second (Table tilt) and the third games being the next easiest, while on the Kinect, games three, four and five were likewise the easiest to improve in.

The children in the Wii group indicated the Ski jump as their favorite, while the children in the Kinect group liked the River rush best.

### 3.3. Intensity of Training

Heart rate data in the first week were only available for 57 children (18 children of the Wii group and 39 children of the Kinect group) due to problems with the heart rate monitor. In the third and final week, we recorded HR data of 33 and 31 children in the Wii group and 38 and 39 in the Kinect group, respectively.

Overall, the estimated maximal heart rate (EMHR) was 195.4 (±0.98) and 60% of the maximal HR was 117.3 (±0.59). Peak HR reached at least 60% of the estimated max HR for 94.7% of the children in the first week, 95.8% in the third and 97.1% in the fifth week. In all three HR readings, 88.8% of the Peak HR values were above the 60% level, 66% of the children had a Peak HR above the 70% level and 26.4% above the 80% of estimated peak HR level. This indicates that at least two third of the recorded children trained above the 70% intensity level during the time it was measured.

The mean rest HR was not different between the Wii and Kinect group (*t*(1,25.1) = −1.98; *p* = 0.06). However, the Mean and Peak heart rate were different between the Wii and Kinect group during training (Figure 4; Mean HR *t*(1,29.6) = −4.93; *p* < 0.001; Peak HR *t*(1,36.5) = −7.54; *p* < 0.001), with the Kinect group reaching higher values.

All frequencies of children reaching the 60, 70 and 80% of the estimated maximum heart rate (EMHR) are significantly different (*p* < 0.01) between the Wii and the Kinect. The HR of the children in the Kinect group showed that they trained on a higher level and only a few children reached the 80% level when training on the Wii (Figure 5).

The mean rest HR over the three moments was not different between TD and DCD (*t*(1,50.9) = −1.52; *p* = 0.14), with a mean of 88.3 ± 10.3 and 92.8 ± 11.3 beats per minute (BPM) respectively. Peak heart rate was also not different between the TD and DCD group, with a mean of 157.1 ± 19.0 and 156.6 ± 16.8 beats per minute (BPM), respectively.

### 3.4. Perceived Exertion

The perceived exertion on the OMNI scale on a scale of 0–10 was not different between the children who played on the Wii compared to the Kinect group (*t*(74) = −0.847, *p* = 0.400) and was very low (mean Wii 1.34 (SD 1.4); mean Kinect 1.61 (1.4)). Likewise, no difference was found between the TD and DCD group (*t*(74) = −0.118, *p* = 0.906; mean for TD 1.50 (SD 1.37) and for DCD 1.46 (SD 1.38)).

### 3.5. Enjoyment

All children highly enjoyed playing the games over the five weeks and this was not dependent on the console. The majority classified their training as ‘Awesome’ (83; 84; 93; 92; 90; 87; 87; 88 and 88% respectively over the nine sessions) and ‘Good fun’ (11; 13; 7; 5; 9; 12; 13; 12 and 11% respectively over the nine sessions) and only a few children scored ‘A bit of fun’ (4 in session 1; one in session 2; 4; 5 and 9). Occasionally, one of the children rated the training as ‘Not so fun’ (in session 2 of the Kinect group) and one in session 1 and 6 of the Wii group as ‘Very boring’.

## 4. Discussion

The purpose of this study was to compare changes in game performance and intensity of HR between the Wii and Kinect training in children with and without DCD. Additionally, the perceived exertion and enjoyment after each training session was monitored per console in children with and without DCD. A significant difference was found between the normalized game scores of the Wii and Kinect, indicating that the children in the Kinect group scored on a higher percentage of the maximum score over the whole training period. However, the rate of change was comparable on both consoles and no difference was found between TD and DCD children. This indicates that both children with and without DCD can improve comparably on both AVG protocols. Playing the Kinect games resulted in higher mean and peak values of HR, and this was not different between children with and without DCD. However, the perceived exertion after playing was very low on both consoles and the level of enjoyment was similarly and mostly rated as ‘Awesome’.

### Kinect and Wii

Our results indicate that the percentage of normalized scores compared to the maximum score is different between the consoles. Apparently, the logarithm of calculating scores for the two consoles is different, starting on a lower level on the Wii Fit and only gaining more points after reaching a higher level of success. This may indicate that training on this type of console can be used more easily for a longer period of training while progress can still be monitored. Remarkably, the majority of the children improved on the games, even though the length of our training period was only five weeks, which indicates that children in this age group easily understood how to play the games and improve their scores. Both AVGs systems provide a degree of autonomy and competence for children, which may increase the feeling of effectively mastering the challenging and fun tasks after they succeed in achieving the desired scores.

The steepness of the performance curve was not different between the Wii and Kinect. In addition, the steepness was not different between the TD and DCD groups. Apparently, controlling the task by shifting weight on the balance board (the Wii system) was not easier for children with DCD, compared to the movements required for the Kinect that demand jumping, bending, kicking or hitting. It is known that children with DCD, after training on the Wii Fit, showed similar rates of improvement compared to their peers [20,21]. This establishes that, despite the differences in motor performance in children with DCD, the games used in the training benefit them, with significant improvements observed over sessions. The results of our study seem to indicate that both AVG systems, with immediate feedback for each movement made, help children with DCD to learn relatively quickly how to play the game on a higher level and improve as much as their peers. Along with previous literature [18,19,20,21], our study highlights the benefits of training using AVGs in children with DCD, also finding Xbox Kinect as effective as Nintendo Wii.

When looking more closely to the types of games, all children learned best on the ‘soccer heading’ and ‘ski jump’ on the Wii. The ‘soccer heading’ challenges a fast but timed shift of the head and trunk to a specific stimulus (ball and avoid other obstacles) and the ‘ski jump’ requires a squat movement with a fast but very timed extension of the legs. The table tilt proved easier for the children with DCD, while TD children learned the ‘balance bubble’ faster. However, both games require a weight shift in all directions with a delicate dosage of this shift, and when moving too far or fast, the bubble will pop or the marble will fall off. The characteristics of both games seem very similar, which makes this difference difficult to understand, with the exception of the popping of the bubble ending the game, while the marble falls on the lower turning table and challenges the player to continue. Apparently, the children with DCD were less able to roll the bubble forward in meters, but could still roll the marble further into holes, since the game continued until the time was finished. The ‘20,000 leaks’ game on the Kinect requires fast movements in all directions in response to sudden leaks which are closed by movements of the hands and feet, while the ‘reflex ridge’ needs fast responses to avoid obstacles by bending, jumping or stepping sideways and the ‘space pop’ requires flapping of the arms and moving the whole body to catch as many bubbles as possible. A common element in these games is the stimuli that are shown first, giving a short time to anticipate and subsequently respond in a timely way. The ‘rally ball’ (Kinect) was more difficult to improve in, since the individual serves a ball and subsequently needs to switch from movements of the head, to one of the arms or legs to hit the fast-returning ball precisely. The predictable setting of the Ski jump was most often the favorite of the Wii games, while the adventurous ‘river rush’ was most often the favorite of the Kinect games. The latter was not the game in which the DCD group improved most easily in contrast with the TD children, but apparently the success rate is not the only factor that determines the preference.

Even though the Wii training setting is more controlled than the Kinect and enhances easier adaptation of the body to control the avatar in the games, which is thought to facilitate improvements for children, there was no difference found between the increase in game performance. However, the Kinect is known to elicit far more free body movements and have the children train under a higher intensity, when considering the heart rate as an indicator. Significantly higher mean and peak values of HR in Kinect boxing were found in comparison with Wii boxing in adults, so the Kinect game is classified as a vigorous-intensity activity [31]. These findings of Sanders and colleagues [31] were supported by our heart rate outcomes presenting a significant difference between the Kinect and Wii games during which at least half of the group of children of the Kinect group reached 80% of the estimated maximal HR during the training. However, Scheer et al. [32] found no significant differences in the HR among Wii Boxing, Kinect Boxing and Sony PlayStation Move and none of the players of the games reached the intensity level of at least moderate intensity [32]. This is why we have to compare our study with other studies with caution. Both studies were carried out with young students during a single short session of 10 min [31] or eight minutes [32]. In contrast, our study lasted nine sessions of 20 min each and HR was registered during three of the training sessions. Additionally, previous studies [33,34,35] also attested Xbox Kinect as a feasible option for children in order to meet moderate to vigorous physical activity recommendations. Based on our results, which console is preferred depends on the aim of training (motor control or health related physical activity). Overall, it might be interesting to offer training using both consoles alternately, to ensure both processes of motor control and physical activity are addressed.

Finally, it is interesting to detect that even though children reached higher intensity levels on the Kinect games, they perceived similar exertion compared to the lower intensity level reached on the Wii games. Apparently, fun and virtual challenges are important factors to consider when offering activities that require energy. Since both the perceived exertion and the enjoyment were similar and, importantly, the enjoyment did not decrease during the 5 weeks, both consoles seem sufficient to offer training during a medium-to-long period of time.

Our study is the first study that directly compared training on two different consoles and compared not only TD children on the Wii and Kinect games, but also children with DCD. It is important to disentangle motor behavior monitored during training in different types of children and this may help physical and occupational therapists and PE teachers to develop or execute extra training that addresses motor control and/or health related physical activity, in addition to real time practice for children with motor problems.

Unfortunately, studies depend on material, and once the material ceases to work sufficiently, as in our Polar HR system, the training cannot be repeated in study designs with children. The insufficiency of material could only be noticed afterwards. Therefore, the heart rate data of the first week must be considered with caution. However, the following occasions presented similar data, which indicated that we still obtained a good impression of the HR values that were reached.

We have accumulated strong knowledge about the effects of AVG on motor learning in children with DCD over the last few years. This comparative AVG study added a novelty, since the Kinect has been less explored than the Wii with this population. However, testing the effects of AVG on physiological responses, such as HR in children with DCD, is a topic that needs more thorough study. It was found that children with DCD benefited from the experiment, but we are aware that the technology is not enough to solve all the issues raised in this study, in which motor and physiological responses are combined.

## 5. Conclusions

It is easier to reach a higher normalized game score of the maximum game score over the whole training on the Kinect compared to the Wii. However, playing on both consoles results in a comparable level of improvement over time. This was not different between DCD and TD groups. Playing the Kinect games resulted in higher mean and peak values of the HR for children with and without DCD. Still, the perceived exertion after playing was very low on both consoles and the level of enjoyment was similarly rated as very high over all sessions, which offers a therapeutic additional tool to train children with DCD over a longer period of time successfully.

## Figures and Tables

**Figure 1 children-09-01823-f001:**
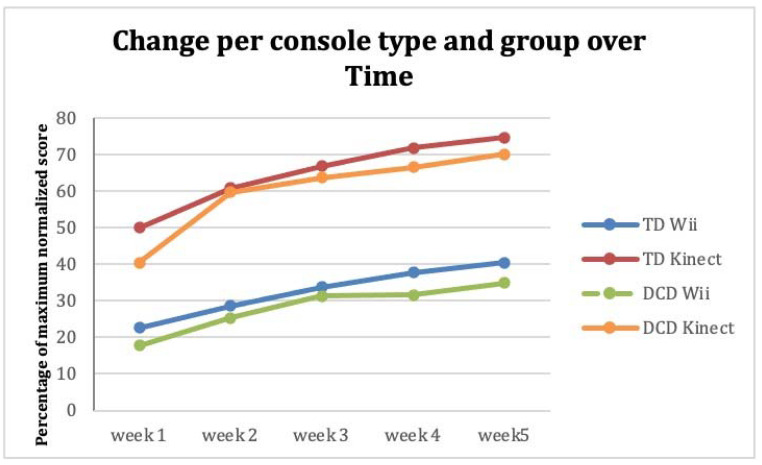
Performance in mean percentage of normalized game scores over five weeks between Wii Fit and Xbox Kinect per participant group.

**Figure 2 children-09-01823-f002:**
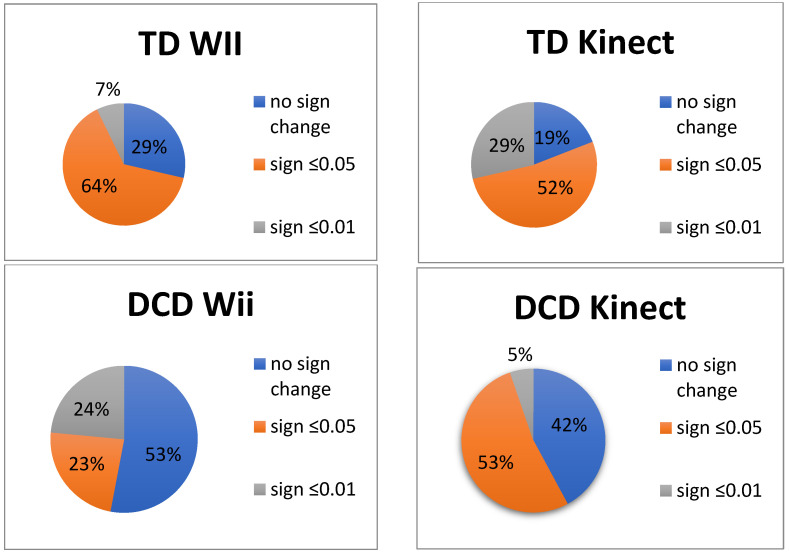
Percentage of individual classification of rate of improvement in based on the best score of the five games played ten times per console per participant group, indicating the percentage of children with no significant change and significant change at ≤0.05 or ≤0.01 level.

**Figure 3 children-09-01823-f003:**
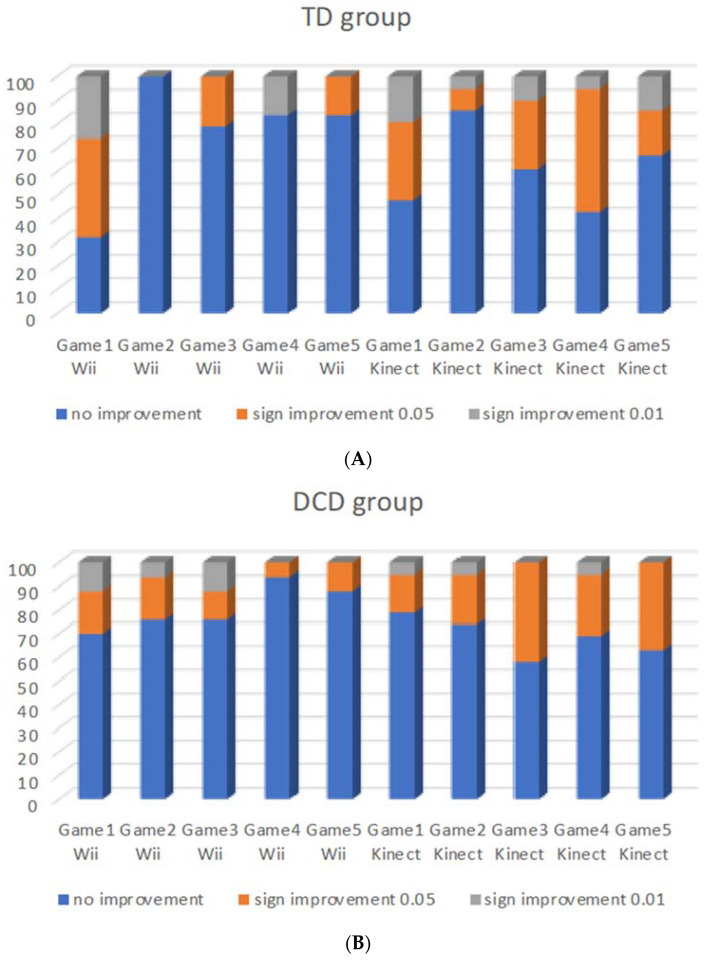
(**A**) (TD group) and (**B**) (DCD group). Individual classification of level of significant improvement within each game over five weeks in the TD group per console (Game 1 Wii = soccer heading; Game 2 Wii = Table tilt; Game 3 Wii = Ski jump; Game 4 Wii = Balance bubble; Game 5 Wii = Penguin slide; Game 1 Kinect = River rush; Game 2 Kinect = Rally ball; Game 3 Kinect = 20,000 leaks; Game 4 Kinect = Reflex ridge; Game 5 = Space pop).

**Figure 4 children-09-01823-f004:**
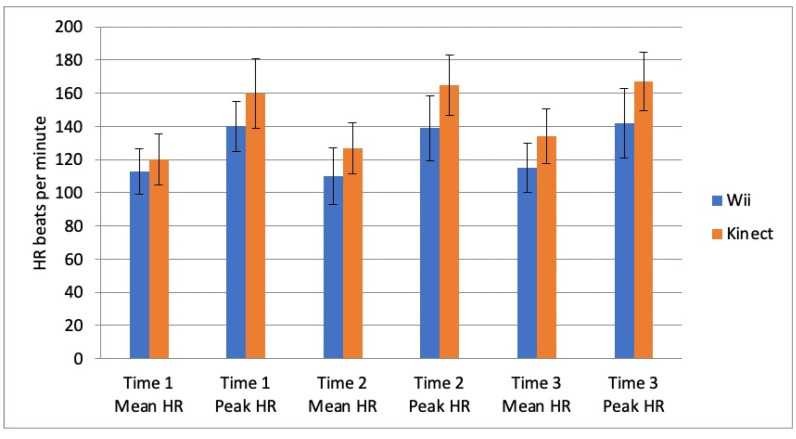
Mean and Peak Heart Rate (HR) recorded in the children in the first, third and fifth week of training on the Wii or Kinect.

**Figure 5 children-09-01823-f005:**
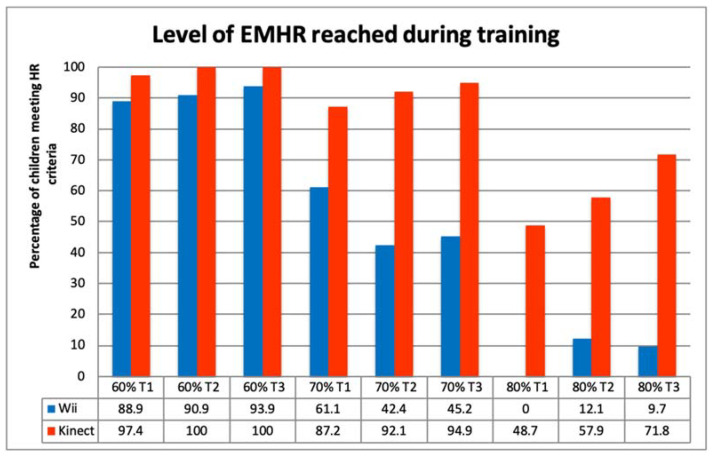
Percentage of children meeting the 60, 70 and 80% level of their estimated maximum heart rate.

**Table 1 children-09-01823-t001:** Descriptive data of participants between Wii and Kinect groups.

Variables	Wii Group (*n* = 36) DCD (*n* = 16) TD (*n* = 20)	Kinect Group (*n* = 40) DCD (*n* = 19) TD (*n* = 21)	*p* Value
Age (years)	9.8 (SD 1.3)	9.7 (SD 1.0)	0.726
Weight (kg)	33.12 (SD 9.04)	35.65 (SD 12.15)	0.953
Height (cm)	140.22 (SD 7.60)	137.96 (SD 10.24)	0.806
BMI	17.21 (SD 1.93)	20.40 (SD 4.72)	0.936
MABC-2	7.31 (SD 2.9)	7.43 (SD 3.1)	0.863
Gender			
Boys Girls	*n* = 20 (55.6%) *n* = 16 (44.4%)	*n* = 22 (55.0%) *n* = 18 (45.0%)	0.358

## Data Availability

Not applicable.

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
