# Peer review of "Active Video Games Performance and Heart Rate on the Wii or Kinect in Children with and without Developmental Coordination Disorder"

_children, 2022, doi:10.3390/children9121823_

Round 1

Reviewer 1 Report

Easy to read and profoundly explained study.

It is unclear why (perceived) exercise rate is measured, as the main aim seems to be to enhance motor learning. It get's more clear in the discussion, and deserves this attention in the introduction.

More specifications are required for inclusion of children in the DCD and in the TD group. If children are randomly approached, much more children could be expected in the TD group. As this is not the case, some additional actions may have been performed to find children for the DCD group, or children might be randomly not included in the TD group to create a comparable number in both groups.

To me, it is not fully clear why best scores on games were used instead of, for example, average scores. Furthermore, the choice for 10 (or 9) different games is unexplained (training one specific game for a longer period of time could also be worthwhile) - and data for the separate games are worth adding to the results section.

Results are clearly described. In Figure 2, however, a description for the gray pieces in the charts is lacking.

More detailed inspection of Hart Rate difference for the TD and DCD group  are suggested as children in the DCD group have a higher HR at rest, and a lower peak HR. Could we learn from this?

Limitations could be more profound as well as the conclusion: what have learned from this study; how might children with DCD (in the end) get better by this knowledge?

Author Response

Children

Special Issue “Impact of Developmental Coordination Disorder on Children”

Prof. Dr. Daniela Pia Rosaria Chieffo

Guest Editor

We thank the editor for the opportunity to adapt the paper according to the feedback we received from the reviewers. By adding the information requested we believe that the paper improved accordingly. Please find below (in red) the itemized responses to the issues raised by the reviewers.

#Reviewer 1:

Comment: Easy to read and profoundly explained study.

Answer: We thank the reviewer for the compliments and the suggestions to improve the paper.

  1. Comments: It is unclear why (perceived) exercise rate is measured, as the main aim seems to be to enhance motor learning. It get's more clear in the discussion, and deserves this attention in the introduction.

Answer: We have added some information about the importance of how intense children perceive the exercise and the fact children with DCD do not perceive it as too exhausting and as fun. We added line 59-61 (Introduction section, page 2): Besides the level of enjoyment, also the level of exertion may be a factor that help children with DCD to remain motivated when interventions extend over a longer period of time. This was added as secondary objective in the abstract as well “Additionally, we assessed the level of improvement per game as well as the perceived exertion and enjoyment during training”.

  1. More specifications are required for inclusion of children in the DCD and in the TD group. If children are randomly approached, much more children could be expected in the TD group. As this is not the case, some additional actions may have been performed to find children for the DCD group, or children might be randomly notincluded in the TD group to create a comparable number in both groups.

Answer: We thank you for this insightful comment about our randomization process to include children in the groups. We agree that much more children are expected in the TD group, but since we conducted the procedures at the school facilities with limited resources we prioritized the children who had the highest scores on the DCDQ to evaluate them with MABC-2, in order to follow the four DSM-5 diagnostic criteria for DCD. We matched them in pairs by age and sex before the randomization with TD children. We have added the requested information as follows: “Based on the DCDQ cut-off scores (≤46 from 5 to 7 years and 11 months; ≤55 from 8 to 9 years and 11 months; and ≤57 from 10 to 15 years old) [33], children with reported motor problems were invited to participate in the study and matched them with children from the same classes whose parents reported no motor problems. (Methods section, Page 3, Lines 87-90). To be clear, we randomly approached them with information letters to be part of the study and all children within the age range in each class of both schools received a sealed envelope containing a letter explaining the procedure and two questionnaires (Developmental Coordination Disorder Questionnaire - DCDQ and a demographic questionnaire) as well as the Informed Consent Form to deliver to their parents.

  1. Comment: To me, it is not fully clear why best scores on games were used instead of, for example, average scores.

We decided to use the best scores of the games, if games were played twice since this score is usually more stable. The Kinect games of sport were played only once due to the length of the games. We preferred to keep the time on task playing AVGs equal for both devices. 

  1. Comment: Furthermore, the choice for 10 (or 9) different games is unexplained (training one specific game for a longer period of time could also be worthwhile) - and data for the separate games are worth adding to the results section.

Answer.  We made a choice for games that were as comparable as possible between the two consoles. As the reviewer will know studies suggest that retention of motor learning is best accomplished with variable training schedules, therefore we introduced task variability (Krakauer 2006).  Not only more repetitions but also more effort leads to better outcome. Importantly, for optimal motor learning we need full engagement of the children. Some children like one game better than the other and we wanted to avoid the children to get bored. This assumption was supported by the fun scores. This is worthwhile particularly for children with DCD, who are known in the literature as less motivated for continuing practice (Katartzi & Vlachopoulos, 2011).

Krakauer JW. Motor learning: its relevance to stroke recovery and neurorehabilitation. Curr Opin Neurol. 2006 Feb;19(1):84-90. doi: 10.1097/01.wco.0000200544.29915.cc. PMID: 16415682.

Katartzi ES, Vlachopoulos SP. Motivating children with developmental coordination disorder in school physical education: the self-determination theory approach. Res Dev Disabil. 2011 Nov-Dec;32(6):2674-82. doi: 10.1016/j.ridd.2011.06.005. Epub 2011 Jul 13. PMID: 21742467.

  1. Comment: Results are clearly described. In Figure 2, however, a description for the gray pieces in the charts is lacking.

Answer: Thank you for pointing this out. The explanation of the gray zone was depicted in the original figures but disappeared when the format was shrunk. The adjustments were made in the manuscript as well.

  1. Comment: More detailed inspection of Hart Rate difference for the TD and DCD group are suggested as children in the DCD group have a higher HR at rest, and a lower peak HR. Could we learn from this?

Answer: The reviewer may have confused difference between the two consoles (Wii/Xbox), which led to differences in HR. There were no significant differences between TD and DCD.

“The mean rest HR over the three moments was not different between TD and DCD (t(1,50.9)= -1.52; p =0.14) with a mean of 88.3 ± 10.3 and 92.8 ± 11.3, beats per minute (BPM) respectively. Peak heart rate was also not different between the TD and DCD group with a mean of 157.1 ± 19.0 and 156.6 ± 16.8, beats per minute (BPM), respectively. “

We have adapted the following two paragraphs and combined them in the discussion to make this finding clearer:

“Even though the Wii training setting is more controlled than the Kinect, and enhances easier adaptation of the body to control the avatar in the games, which is thought to facilitate improvements for children, there was no difference found between the increase in game performance. However, the Kinect is known to elicit far more free body movements and have the children train under a higher intensity, when considering the heart rate as an indicator. Significantly higher mean and peak values of HR in Kinect boxing were found in comparison with Wii boxing in adults classified Kinect as a vigorous-intensity activity [35]. These findings of Sanders and colleagues [35] were supported by our heart rate outcomes presenting a significant difference between the Kinect and Wii games during which at least half of the group of children of the Kinect group reached 80% of the estimated maximal HR during the training. However, Scheer, et al. [36] found no significant differences in the HR among Wii Boxing, Kinect Boxing and Sony PlayStation Move and none of the players of the games reached the intensity level of at least moderate intensity [36]. This is why we have to compare our study with other studies with precaution. Both studies were carried out with young students during a single short session of 10 minutes [35] or eight minutes [36]. In contrast, our study lasted nine sessions of 20 minutes each and HR was registered during three training sessions. Additionally, previous studies [37–39] also attested Xbox Kinect as a feasible option for children in order to meet moderate to vigorous physical activity recommendations. Based on our results it depends on the aim of training (motor control or health related physical activity) which console is preferred. Overall, it might be interesting to offer a training using both consoles alternately, to ensure both processes of motor control and physical activity are addressed (Discussion section, Page 12, Lines 361-376).

  1. Comment: Limitations could be more profound as well as the conclusion: what have learned from this study; how might children with DCD (in the end) get better by this knowledge?

Answer: we have added the following sentence to the strength and limitations as follows:

We have accumulated strong knowledge about the effects of AVG on motor learning in children with DCD over the last few years. This comparator AVG study added a novelty since Kinect has been less explored than Wii with this population. However, testing the effects of AVG on physiological responses, like HR with children with DCD is a topic that needs more thorough study. It was found that children with DCD might be benefited from the experiment conducted in this study, but we are aware that the technology is not enough to solve all the issues raised in this study, in which motor and physiological responses are combined. (Discussion section, Pages 12-13, Lines 394-400).

In addition, we have extended the conclusion with another recommendation how the findings could be used for children with DCD in clinical practice. As follows: Still, the perceived exertion after playing was very low on both consoles and the level of enjoyment was similarly rated as very high over all sessions, which offers a therapeutic additional tool to train children with DCD over a longer period of time successfully (Conclusion section, Page 13, Lines 407-409).

#Reviewer 2:

Thanks for being invited to review the paper, and few comments are as follows:

We appreciate the reviewer for the compliments and comments raised.

  1. Comment. Why no group difference in MABC2 total standard score between the Wii or Kinects groups? 

Answer: We had no differences on the pre-test because we matched them in pairs before the randomization. We have added extra information about the selection of the children as follows: Based on the DCDQ cut-off scores (≤46 from 5 to 7 years and 11 months; ≤55 from 8 to 9 years and 11 months; and ≤57 from 10 to 15 years old) [33], children with reported motor problems were invited to participate in the study just like the children from the same classes whose parents reported no motor problems.

  1. Comment: Why no within-group differences in MABC2 total scores between Will DCD/TD and Kinect DCD/TD group? 

Answer: We had no differences on the pre test because we matched them in pairs (DCD and TD children together in the Wii group and the same combination in the Kinect group) before the randomization. We have added extra information about the selection of the children (see answer comment 1).

  1. Comment: Are the OMNI scale and Enjoyment Scale developed by authors? could you please provide psychometrics of both measures? 

Answer: The Children's OMNI scale of perceived exertion (OMNI Scale) has been validated for children, attesting good correlation values (0.85-0.94) in relation to the total body oxygen uptake (Robertson et al. 2004). The Enjoyment Scale was developed for our earlier studies (Jelsma et al. 2014)

Robertson RJ, Goss FL, Dubé J, et al. Validation of the Adult OMNI Scale of Perceived Exertion for Cycle Ergometer Exercise. Med Sci Sports Exerc. 2004;36:102–108.

Jelsma D, Geuze RH, Mombarg R, et al. The impact of Wii Fit intervention on dynamic balance control in children with probable Developmental Coordination Disorder and balance problems. Hum Mov Sci [Internet]. 2014;33:404–418. Available from: http://dx.doi.org/10.1016/j.humov.2013.12.007.

Comment: Could you please provide descriptive data of participants? 

Answer: we summarized the descriptive data in table 1 as follows:

Table 1. Descriptive data of participants between Wii and Kinect groups

Variables

Wii group (n=36)

DCD (n=16)

TD (n=20)

Kinect group (n=40)

DCD (n=19)

TD (n=21)

p-value

Age (years)

9.8 (SD 1.3)

9.7 (SD 1.0)

.726

Weight (kg)

33.12 (SD 9.04)

35.65 (SD 12.15)

.953

Height (cm)

140.22 (SD 7.60)

137.96 (SD 10.24)

.806

BMI

17.21 (SD 1.93)

20.40 (SD 4.72)

.936

MABC-2

7.31 (SD 2.9)

7.43 (SD 3.1)

.863

Gender

Boys

Girls

n=17 (44.7%)

n=21 (55.3%)

n=22 (55.0%)

n=18 (45.0%)

.358

Reviewer 2 Report

Thanks for being invited to review the paper, and few comments are as follows:

1. Why no group difference in MABC2 total standard score between the Wii or Kinects groups? 

2. Why no within-group differences in MABC2 total scores between Will DCD/TD and Kinect DCD/TD group? 

3. Are the OMNI scale and Enjoyment Scale developed by authors? could you please provide psychometrics of both measures? 

4. Could you please provide descriptive data of participants? 

Author Response

(The authors gave the same response as above.)
